# Loss of Microglial Insulin Receptor Leads to Sex-Dependent Metabolic Disorders in Obese Mice

**DOI:** 10.3390/ijms23062933

**Published:** 2022-03-08

**Authors:** Irina V. Milanova, Nikita L. Korpel, Felipe Correa-da-Silva, Eline Berends, Samar Osman, Susanne E. la Fleur, Eric Fliers, Andries Kalsbeek, Chun-Xia Yi

**Affiliations:** 1Department of Endocrinology and Metabolism, Amsterdam University Medical Centers, University of Amsterdam, 1105 AZ Amsterdam, The Netherlands; i.v.milanova@amsterdamumc.nl (I.V.M.); n.korpel@nin.knaw.nl (N.L.K.); f.correadasilva@amsterdamumc.nl (F.C.-d.-S.); e.berends@maastrichtuniversity.nl (E.B.); samar.osman97@live.nl (S.O.); s.e.lafleur@amsterdamumc.nl (S.E.l.F.); e.fliers@amsterdamumc.nl (E.F.); a.kalsbeek@nin.knaw.nl (A.K.); 2Laboratory of Endocrinology, Amsterdam University Medical Centers, Amsterdam Gastroenterology Endocrinology Metabolism (AGEM), 1105 AZ Amsterdam, The Netherlands; 3Netherlands Institute for Neuroscience, Royal Netherlands Academy of Arts and Sciences, 1105 BA Amsterdam, The Netherlands

**Keywords:** microglia, obesity, immunometabolism, diabetes, sex difference

## Abstract

Obesity and type 2 diabetes mellitus (T2DM) are highly prevalent disorders, associated with insulin resistance and chronic inflammation. The brain is key for energy homeostasis and contains many insulin receptors. Microglia, the resident brain immune cells, are known to express insulin receptors (InsR) and to be activated by a hypercaloric environment. The aim of this study was to evaluate whether microglial insulin signaling is involved in the control of systemic energy homeostasis and whether this function is sex-dependent. We generated a microglia-specific knockout of the InsR gene in male and female mice and exposed them to control or obesogenic dietary conditions. Following 10 weeks of diet exposure, we evaluated insulin tolerance, energy metabolism, microglial morphology and phagocytic function, and neuronal populations. Lack of microglial InsR resulted in increased plasma insulin levels and insulin resistance in obese female mice. In the brain, loss of microglial InsR led to a decrease in microglial primary projections in both male and female mice, irrespective of the diet. In addition, in obese male mice lacking microglial InsR the number of proopiomelanocortin neurons was decreased, compared to control diet, while no differences were observed in female mice. Our results demonstrate a sex-dependent effect of microglial InsR-signaling in physiology and obesity, and stress the importance of a heterogeneous approach in the study of diseases such as obesity and T2DM.

## 1. Introduction

Obesity and type 2 diabetes mellitus (T2DM) are global chronic disorders mainly caused by a disbalance between food intake and energy expenditure [1]. In the central nervous system (CNS), the hypothalamus is the key region responsible for maintaining energy homeostasis, containing neurons known to sense nutrients as well as circulating hormones [2,3,4,5,6]. Insulin receptors (InsR) are broadly expressed in the brain, with high concentrations in various regions, including the olfactory bulb, cerebral cortex, hippocampus, and hypothalamus [7]. Many studies have confirmed the importance of brain InsR-signaling in the regulation of food intake and energy expenditure [8,9,10]. Insulin affects energy homeostasis by promoting the activity of the anorexigenic proopiomelanocortin (POMC) neurons and inhibiting the activity of the agouti-related peptide/neuropeptide Y (AgRP/NPY) orexigenic neurons, thus promoting energy expenditure [11,12,13,14].

A key factor known to disrupt energy homeostasis and cause obesity is a hypercaloric diet [15]. Obesity is associated with low-grade chronic peripheral inflammation [16,17,18]. It has been shown that dietary-induced obesity (DIO) in rodents is a potent microglial activator in the hypothalamus [19,20,21,22,23,24]. Microglia are the innate immune cells in the CNS, which play a key role in maintaining a healthy neuronal environment [25]. Interestingly, microglial cells also express InsRs, suggesting a direct effect of insulin on microglia [26]. A recent study investigating the role of insulin in a murine microglial cell line (BV2) found that insulin decreased the innate immune response and altered the phagocytic capacity of microglia [27]. Yet, the precise role of microglial insulin signaling in physiological and obesogenic conditions remains largely unknown.

We hypothesized that reduced microglial insulin signaling affects the microglial immune function, which will ultimately result in insufficient maintenance of a healthy brain environment and neuronal dysfunction. To test our hypothesis, we generated mice with a microglia-specific knockout of the InsR gene. The aim of this study was to evaluate whether microglial insulin signaling is involved in the control of systemic energy homeostasis. Since it has been shown that estrogen may have a protective effect on the development of insulin resistance, diabetes, and obesity, both in humans and animals [28], we performed our studies in both male and female mice to assess whether gender plays a role in the progression of obesity in control and microglial InsR-deficient mice.

## 2. Results

### 2.1. Microglial InsR Deletion in Mice Affects Food and Water Intake during Dietary-Induced Obesity in a Sex-Dependent Manner

To evaluate the role of InsR function in microglia in obesity, we used the Cre/loxP method to specifically remove the InsRs from microglial cells. Mice carrying lox-P flanked Insr alleles (Insr^fl/fl^) were crossed with mice carrying tamoxifen-inducible Cre recombinase, driven by the Cx3cr1 promoter (Cx3Cr1^CreERT2^) (Figure 1A). We used primary microglial cells from Insr^fl/fl^—Cx3Cr1^CreERT2^ mice to generate InsR knockout microglia (InsR-KO, mixed gender) in vitro. Primary microglia from Insr^wt/wt^—Cx3Cr1^CreERT2^ were used as controls (InsR-Ctrl). All primary microglial cells were treated with 4-hydroxytamoxifen to delete the Insr gene. We confirmed a significant decrease of InsR protein in InsR-KO microglia compared to InsR-Ctrl (Figure 1B,C).

To understand the importance of microglial InsR function in vivo, we induced the knock-out as mentioned above at the adult stage of mice. Both male and female mice were treated with tamoxifen at 6 weeks of age to generate microglial InsR-Ctrl and InsR-KO animals. At 8 weeks of age, mice were fed either a Chow diet, representing the physiological dietary environment or high-fat diet (HFD), known to induce obesity, for the duration of 12 weeks. Both male and female microglial InsR-KO mice, fed an HFD, showed increased caloric intake at the end of 12 weeks of HFD exposure when compared to HFD-fed InsR-Ctrl mice (Figure 1D,F). Interestingly, female HFD-fed InsR-KO mice showed higher caloric intake after 3 weeks of HFD exposure, compared to female HFD-fed InsR-Ctrls (Figure 1D), while male InsR-KO mice had lower caloric intake after 6 and 9 weeks of diet exposure compared to InsR-Ctrl male mice under obesogenic conditions (Figure 1F). Both male and female HFD-fed InsR-KO mice had lower water intake at week 3 and higher water intake at week 12, when compared to InsR-Ctrl mice on HFD (Figure 1E,G).

When evaluating the effect of microglial loss of InsR function in females on body weight, we found no significant differences in the BW and BW gain between InsR-Ctrl and InsR-KO mice on the Chow diet. However, we observed a strong trend for increased BW and BW gain from week 3 onwards in obese InsR-KO females, compared to InsR-Ctrl (Figure 1H,I). In males, we observed no difference between InsR-KO and InsR-Ctrl male mice during the progression of obesity (Figure 1H–K). Both InsR-KO and InsR-Ctrl mice fed a HFD showed increased BW and BW gain compared to their respective controls (Figure 1H–K).

While evaluating the effect of microglial InsR-KO on adiposity, we found that Chow-fed InsR-Ctrl female mice had lower perirenal white adipose tissue (WAT), compared to InsR-KO mice on Chow diet, as well as compared to InsR-Ctrl, HFD-fed mice (Figure 1L). No differences were observed in female gonadal WAT between any of the groups (Figure 1M). In males, both InsR-KO and InsR-Ctrl mice fed an HFD showed increased BW and BW gain, as well as perirenal and epididymal WAT, compared to their respective controls (Figure 1H–K). InsR-KO males had higher perirenal WAT, compared to InsR-Ctrl on Chow diet (Figure 1N). We observed no differences in adiposity between InsR-KO and InsR-Ctrl animals fed an HFD in both male and female mice (Figure 1L–O).

These data suggest that loss of microglial InsR may play a role in the progression of obesity in a sex-dependent manner.

### 2.2. Microglial Insulin Signaling in Female Mice Plays an Important Role in Maintaining Homeostatic Glucose and Insulin Concentrations

Next, we evaluated the effect of microglial insulin function on glycemic homeostasis and insulin sensitivity in both control and obesogenic conditions. Following 11 weeks of obesogenic diet exposure, we observed a decrease in fasting glucose in InsR-KO female mice compared to InsR-Ctrl (Figure 2A). Moreover, HFD-fed female InsR-KO mice showed a failure to respond to insulin following an insulin tolerance test when compared to InsR-Ctrl mice, as seen in higher glucose levels and a smaller net area under the curve (AUC) (Figure 2B,C). Additionally, InsR-KO females had higher plasma insulin levels on HFD, compared to the Chow diet (Figure 2D). We observed no differences in basal blood glucose between InsR-KO and InsR-Ctrl female mice prior to the start of HFD-feeding (data not shown).

In males, InsR-KO mice had higher basal blood glucose compared to InsR-Ctrl mice one week following tamoxifen injection (Appendix A). However, that difference was not maintained during the study. We found no differences in glucose values following an insulin bolus (Appendix A), and there were no changes in fasting plasma insulin (Appendix A) between InsR-KO an InsR-Ctrl male mice on either diet. Both InsR-Ctrl and InsR-KO male mice had higher fasting plasma insulin on HFD, compared to the Chow diet (Appendix A).

These data confirm our sex-dependent observations and suggest a key role of microglial insulin signaling in glucose homeostasis in female mice during obesity.

### 2.3. Microglial Insulin Function Has a Sex-Dependent Effect on Energy Metabolism in Obesity

To further investigate the implications of lack of microglial InsR signaling in physiological and obesogenic conditions, we evaluated different metabolic parameters following chronic exposure to control or hypercaloric diet.

Female InsR-KO mice had a lower respiratory exchange rate (RER) during the dark (active) phase when compared to InsR-Ctrl under obesogenic conditions (Figure 2E,F), indicating a higher use of lipids as an energy substrate in HFD-fed InsR-KO females compared to InsR-Ctrl. This difference was not observed when comparing 24h RER (Figure 2G). We observed no differences in locomotor activity (Appendix A) and energy expenditure (EE) (Appendix A) between InsR-KO and InsR-Ctrl female mice.

In males, no differences in RER were observed between InsR-KO and InsR-Ctrl mice, irrespective of the dietary intake (Figure 2H,I). InsR-Ctrl and InsR-KO mice fed a Chow diet had higher RER during the dark (active) phase compared to light, and that difference was lost when the animals were fed an HFD (Figure 2H,I). We observed a lower locomotor activity in HFD-fed, InsR-Ctrl male mice compared to InsR-KO mice during the light (inactive) phase (Figure 2K,L), which was borderline significant for 24 h (Figure 2M). These data could indicate changes in mood regulation as it has been shown that decreased locomotor activity is associated with behavioral mood changes [29,30]. No differences were observed in EE between InsR-KO and InsR-Ctrl male mice, irrespective of diet (Appendix A).

These data suggest that loss of InsR in obese female mice leads to higher reliance on fat metabolism during the active phase of the animals.

### 2.4. Lack of Microglial InsR Has No Effect on Satiety-Involved Hypothalamic Neuropeptides in Physiological and Obesogenic Conditions

Next, we wanted to evaluate the importance of microglial InsR signaling in physiological and obesogenic conditions on key neuronal populations in the hypothalamus involved in satiety—hypocretin (Hcrt), agouti-related peptide (Agrp), and neuropeptide Y (Npy). We observed no changes in *Hcrt*, *Agrp*, and *Npy* gene expression between InsR-KO and InsR-Ctrl animals, irrespective of sex or diet (Figure 3A–F). We found a decrease in *Hcrt* expression in HFD-fed Insr-Ctrl males comapred to the Chow diet. Taken together, these data suggest that loss of microglial InsR does not affect the function of hypothalamic neuropeptides involved in the control of food intake and satiety.

To assess the changes in microglial innate immunity in the hypothalamus, following the loss of microglial InsR, we evaluated the gene expression of key genes involved in microglial function, cluster of differentiation 68 (*Cd68*), a commonly used phagocytic indicator in Iba1-ir microglia; Interleukin-1 beta (*Il1b*) cytokine, highly produced by microglia; and inhibitor of nuclear factor-kappa b kinase subunit beta (*Ikbkb*), kinase involved in the phosphorylation of the inhibitor/NFkB complex, thus activating the nuclear factor kappa-light-chain-enhancer of activated B cells (NFkB), involved in an inflammatory response. We found an increase in *Cd68* gene expression in female InsR-KO mice compared to InsR-Ctrl mice under obesogenic conditions (Appendix A), while no changes were observed in male mice (Appendix A). We found no changes in *Il1b* and *Ikbkb* gene expression between InsR-KO and InsR-Ctrl mice, irrespective of diet or sex (Appendix A), suggesting no direct effect on hypothalamic innate immunity.

### 2.5. Hypothalamic Microglial Morphology and Phagocytic Capacity Are Dependent on Insulin Signaling in Male and Female Mice

To investigate the importance of hypothalamic microglial insulin signaling for immune function during obesity, we evaluated microglial cells in the arcuate nucleus (ARC), a key brain nucleus involved in systemic control of energy metabolism. Microglial cell number and morphology were evaluated by ionized calcium-binding adaptor molecule 1 (Iba1) immunoreactivity. We found a borderline significant decrease of microglial cell number in InsR-KO female mice, compared to InsR-Ctrl during obesogenic conditions (Figure 4A,B). In males, we found no effect of InsR on microglial cell number. However, as expected from previous research, we confirmed higher microglial number in InsR-Ctrl mice when fed an HFD, compared to Chow (Figure 5A,B). Interestingly, we found a decrease in primary microglial projections in all InsR-KO animals compared to InsR-Ctrl animals, irrespective of diet or sex (Figure 4C and Figure 5C), which suggests a prominent effect of insulin signaling on microglial morphology.

As phagocytosis is a key function of microglial cells and we observed changes in *Cd68* gene expression in the whole hypothalamus of female mice, next we evaluated the phagocytic capacity of Iba1-ir cells specifically in the ARC. We evaluated the immunoreactivity of CD68 in Iba1-ir microglia. We found higher CD68-ir in microglia of InsR-KO female mice on Chow diet when compared to InsR-Ctrl mice (Figure 4D,E), indicating a possible involvement of insulin receptor signaling in microglial phagocytosis in females. Male InsR-KO mice had lower CD68-ir in microglia when fed HFD diet, compared to chow diet (Figure 5D,E).

These data suggest an important role of microglial insulin signaling for microglial morphology. Moreover, there seems to be a sex-dependent role of insulin on microglial immune and phagocytic function during health and obesity.

### 2.6. InsR Loss in Microglial Cells Leads to a Decrease of POMC Neurons during Obesity in Male Mice

The ARC nucleus contains the main subsets of neurons involved in energy metabolism. Here we investigated the role of microglial insulin signaling during health and obesity on the anorexigenic proopiomelanocortin (POMC) neurons, as they have been previously shown to be highly sensitive to a chronic hypercaloric environment [31]. We evaluated the total POMC-ir cell number in the ARC. We found no changes in POMC-ir in female InsR-KO and InsR-Ctrl mice on either Chow or HFD (Appendix A). However, in males, we observed a borderline significant increase in the number of POMC neurons in InsR-KO compared to InsR-Ctrl mice when fed a Chow diet (Figure 5F,G). Additionally, InsR-KO male mice had a significant decrease in POMC-ir on HFD, compared to the control Chow diet (Figure 5F,G). These data suggest a role for microglial insulin signaling on POMC neuronal function in male mice.

## 3. Discussion

In this study, we showed that microglial insulin signaling plays a sex-dependent role in the control of systemic energy homeostasis, with microglial InsR being involved in the progression of obesity most prominently in female mice. We found a decrease in primary microglial projections in all InsR-KO animals compared to InsR-Ctrl animals, irrespective of diet or sex. In obese female mice, loss of microglial InsR resulted in decreased microglial cell numbers, higher fasting plasma insulin, and insulin resistance. Therefore, we propose that microglial insulin signaling plays a key protective role during the progression of obesity in female mice.

The decrease in the primary microglial projections in all microglia-InsR-KO animals, when compared to microglia-InsR-Ctrl animals, is in line with the study of García-Cáceres et al., who reported that astrocytes lacking insulin receptors have fewer and shorter primary projections when compared to control astrocytes [32]. In addition, we found a trend for decreased microglial cell number in obese microglia-InsR-KO female mice, compared to obese microglia-InsR-Ctrls, while no differences were observed in males, which could be suggesting a direct protective effect of insulin signaling on microglia in females during obesity. We also observed a sex-dependent effect of microglial insulin function on the phagocytic capacity of the cells. Female microglia-InsR-KO mice showed an increase in phagocytic indicator CD68 compared to microglia-InsR-Ctrl in homeostatic conditions. However, in male mice, we observed no differences in physiological dietary conditions, while microglia-InsR-KO male mice had lower CD68 compared to microglia-InsR-Ctrl males in DIO. Previously it has been shown that microglial phagocytosis is sex-dependent and that estradiol can decrease the phagocytic microglia in rat hippocampus in females [33]. This could give a potential explanation for the sex-dependent difference we observed in microglial phagocytic function. Taken together, these data suggest that defective insulin signaling in glial cells leads to disruption of their morphology. This aberrant morphological trait indicates an impairment of their surveying function and capacity for interaction with other cell types, as well as phagocytic capacity, which could lead to severe impairment of brain homeostasis. Studies have shown that intranasal insulin treatment can reverse microglial activation and improve cognition, which highlights the importance of microglial insulin signaling [34,35,36].

Due to the important role of the hypothalamus, and specifically the ARC nucleus in energy metabolism, we decided to study the effect of microglial insulin receptor loss on the anorexigenic population of POMC neurons. We observed no changes in POMC neurons in females, while there was a significant loss of POMC neurons in microglia-InsR-KO male mice, compared to microglia-InsR-Ctrl in obesity. This sex-specific decrease of POMC neurons probably can be explained by the decrease of phagocytic capacity we found in microglia-InsR-KO males in obesity, as previously it has been demonstrated that impaired microglial phagocytic capacity leads to loss of POMC neurons [37,38]. Additionally, circulating estrogens have been shown to maintain insulin sensitivity in POMC neurons in obese females, which could explain why we found no loss of POMC neurons in microglia-InsR-KO obese females, even though they presented with worsened obesogenic outcomes [39].

Up until recently, rodent studies on metabolic disorders mainly focused on males and sex differences have been understudied, especially concerning microglia. However, emerging data are pointing to a strong heterogeneity of microglia between males and females [40,41]. A study by Dorfman et al., showed that a sex-specific CX3CR1 hypothalamic signaling is at the core of obesity progression, with female mice maintaining a level of resistance to DIO [42]. In a human study, women have been shown to maintain better insulin sensitivity, which could be partially explained by a beneficial effect of estrogen on insulin and glucose homeostasis [28]. Research has shown that estradiol increases IGF-I receptor and Akt phosphorylation, which points to a possible mechanism for the metabolic, neuroendocrine, and neuroprotective effects [43]. Here we report that lack of microglial insulin signaling in obesity impaired insulin sensitivity in female mice, which suggests a protective role of intact microglial InsR function in obese females. This is further confirmed by our observations on fasted plasma insulin, as we found no difference between healthy and obese microglia-InsR-Ctrl female mice, while microglia-InsR-KO obese females had a higher level of insulin, compared to microglia-InsR-KO females, fed a Chow diet.

In summary, the findings of this study highlight the importance of microglial insulin signaling in the CNS control of energy metabolism and the sex-dependent differences observed during the progression of obesity. Microglial cells seem to show a vast difference in their phenotype between males and females in an age-, region-, and disease-dependent manner [40,41,44]. Our findings provide further insight into the complex mechanisms that are operative in the brain for the control of energy metabolism in health and disease and highlight the importance of taking into account possible sex-dependent differences in these mechanisms. Clearly, taking into account gender differences is crucial for the development of effective therapeutics for diseases such as obesity and T2DM in both males and females.

## 4. Materials and Methods

### 4.1. Animals

Microglia-specific postnatal InsR-KO male and female mice were generated by crossing InsR^fl/fl^ mice (#006955, JAX^®^) [45] with Cx3Cr1^CreERT2^ mice (#020940, JAX^®^), which express the tamoxifen-inducible Cre recombinase, driven by the fractalkine chemokine (C-X3-C motif) receptor 1 (Cx3Cr1) promoter [46]. All mice were group-housed on a 12-h-light/12-h-dark cycle (lights on at 7:00 am; Zeitgeber time zero (ZT0)) at 22 ± 2 °C and had access to food and water ad libitum unless stated otherwise. At the age of 6 weeks, the animals were administered 2100 μL intraperitoneal (i.p.) injections of tamoxifen (Ref.: T5648, Merck) 48 h apart at a concentration of 20 mg/mL (dissolved in 90% Sunflower Oil (Ref: S5007, Merck) and 10% pure Ethyl alcohol), to excise exon 4 of the InsR gene, flanked by loxP sites. InsR^fl/fl^-Cx3Cr1^CreERT2^ mice are referred to from now onwards as InsR-KO animals. Their littermates, InsR^wt/wt^-Cx3Cr1^CreERT2^ mice, are referred to as InsR-WT animals (controls). All studies were approved by the Animal Ethics Committee of the Royal Dutch Academy of Arts and Sciences (KNAW, Amsterdam) and performed according to the guidelines on animal experimentation of the Netherlands Institute for Neuroscience (NIN, Amsterdam).

### 4.2. Primary Microglial Cell Culture

Primary microglial cells were isolated from neonatal InsR^wt/wt^-Cx3Cr1^CreERT2^ and InsR^fl/fl^-Cx3Cr1^CreERT2^ mouse brains. Briefly, brains were cultured until 80–90% confluency of mixed glial culture, when cells were provided with L929 cell line-conditioned medium containing colony-stimulating factor-1 (CSF1) to promote microglial proliferation. Microglial cells were collected and seeded for evaluation of knock-out efficiency. All cells were treated with a culture medium containing 1 µM 4-hydroxytamoxifen for 48 h and further used for evaluation of InsR protein.

### 4.3. Western Blot

Isolated primary microglial samples (see above) were used for evaluation of knock-out efficiency by measuring InsR protein concentration, following the established protocol (see supplementary methods).

Primary antibodies: rabbit anti-insulin receptor β (Ref: 3025, Cell Signaling), goat anti-actin (Ref: sc-1616, Santa Cruz). Secondary antibodies: rabbit anti-goat immunoglobulins/HRP (Ref: P0449, Dako), goat anti-rabbit immunoglobulins/HRP (Ref: P0448, Dako).

Data presented consists of a comparison between the gray area in InsR-KO and InsR-Ctrl samples in ImageJ [47].

### 4.4. Metabolic Phenotyping

Basal blood glucose levels of ad libitum fed animals were measured at the age of 7 weeks. Blood was obtained from a puncture in the tail vein. Glucose levels (mmol/L) were measured with a glucometer (FreeStyle, Abbott Diabetes Care). Obesity was induced at the age of 8 weeks with a diet containing 58 kcal% fat and 25 kcal% carbohydrates (HFD, 5.56 kcal/g, D12331, Research Diets inc., New Brunswick, USA), for the duration of 10 weeks. Control animals were fed a standard chow diet (3.1 kcal/g, 2018, Teklad diets, Invigo). Body weight (BW), and cumulative food and water intake were monitored twice per week throughout the whole study. Data presented in this manuscript is the BW gain per week and average food and water intake per animal per week. At 18 weeks of age, i.e., after 10 weeks of HFD exposure, respiratory exchange rate (RER), locomotor activity, and heat production were measured by a customized indirect gas calorimetric system (TSE Systems, GmbH). In short, chow- or HFD-fed mice were single housed in metabolic cages equipped with a PhenoMaster module for indirect gas calorimetry for 3 days for adaptation, followed by a continuous 48-h cycle of measurement of oxygen consumption (VO2) and carbon dioxide production (VCO2), as well as XT activity (all horizontal beam breaks in one direction in counts), and Z activity (all vertical beam breaks in counts) 4 times per hour. These parameters were used for the calculation of the RER, activity, and energy expenditure (EE). After 12 weeks of diet exposure, the animals were sacrificed by euthanasia with 60% CO_2_/40% O_2_, followed by decapitation. Perirenal, epididymal and gonadal white adipose tissues (WAT) were dissected upon sacrifice for evaluation of fat mass gain.

### 4.5. Insulin Tolerance Test

An intraperitoneal insulin tolerance test (ipITT) was performed at 19 weeks of age by injecting an insulin bolus (0.75 U/kg for males, 0.5 U/kg for females solved in saline) (Actrapid HM 100 IU/mL, Novo Nordisk) in 4 h-fasted mice. Tail blood glucose levels (mmol/L) were measured with a glucometer (FreeStyle, Abbott Diabetes Care) before (0 min) and at 20, 30, 45, 60, 90, and 120 min after injection.

### 4.6. Insulin ELISA

Blood plasma was used to evaluate fasted insulin concentration with the Rat/Mouse Insulin ELISA Kit (Ref: EZRMI-13K, Merck), following the manufacturer’s instructions. The kit has inter-assay precision 6.0–17.9% and intra-assay precision 0.9–8.4%, and a lower detection limit 0.2 ng/mL. All samples were measured in duplicate.

### 4.7. PCR

Hypothalamic gene expression was evaluated using quantitative PCR (qPCR). Briefly, RNA was isolated using Tri-reagent (Ref: T9424-200, Sigma) and the ISOLATE II RNA Mini Kit (Ref: BIO-52073, Meridian Bioscience) following the manufacturer’s guidelines. cDNA was synthesized using the Transcriptor First Strand cDNA Synthesis Kit (Ref: 04896866001, Roche Life Science) following the manufacturer’s guidelines. qPCR was performed with the LightCycler^®^ 480 (Roche Life Science) for *Hcrt*, *Agrp*, *Npy*, *Cd68*, *Il1b* and *Ikbkb* genes (for primer sequences, see supplementary methods). Gene expression is presented as a relative fold change to housekeeping genes (*Hprt* and *bActin*).

### 4.8. Immunohistochemical and Immunofluorescent Staining

Animals were sacrificed at 20–21 weeks of age by perfusion fixation (see supplementary methods). Hypothalamic 30 µm thick coronal sections were selected from each mouse covering the rostral–caudal arcuate (ARC) nucleus region. Immunohistochemical staining of the ionized calcium-binding adaptor molecule 1 (Iba1)- and proopiomelanocortin (POMC)-ir were performed to profile, respectively, the microglial morphology and the neighboring POMC neurons; immunofluorescent co-staining of Iba1 and CD68 was performed to examine microglial phagocytic capacity in the ARC region. See supplementary methods for further information on staining protocols.

Primary antibodies: rabbit anti-iba1 (Ref: 234003, Synaptic Systems), rabbit anti-POMC (Ref: H-029-30, Phoenix Pharmaceuticals), rat anti-CD68 (ab53444, Abcam).

Secondary antibodies: goat anti-rabbit IgG biotinylated (Ref: BA-1000, Vector Laboratories), goat anti-rat IgG, biotinylated (Ref: BA-9400, Vector Laboratories). Fluorescent secondary antibodies: Streptavidin, Alexa Fluor 488 (S32354, Invitrogen), goat anti-rabbit IgG, Alexa Fluor 594 (A11037, Invitrogen).

### 4.9. Statistical Analysis

Statistical analyses were performed using Graphpad PRISM (version 8.4.2, GraphPad Software, Inc., San Diego, CA, USA). Statistical significance was determined using one-way, two-way ANOVA and student *t*-test. Data are presented as mean ± SEM. Results were considered statistically significant when *p* < 0.05.

## Figures and Tables

**Figure 1 ijms-23-02933-f001:**
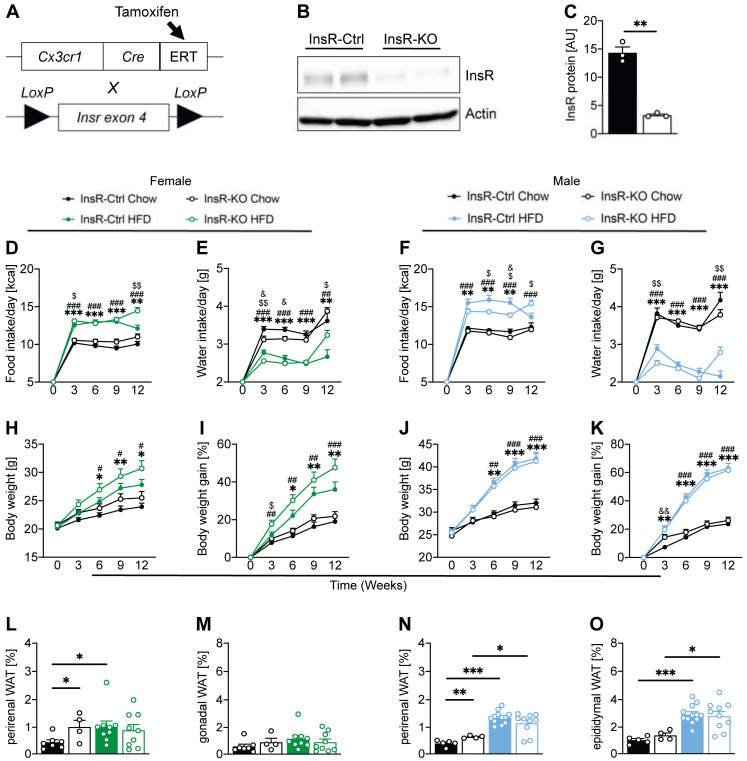
Lack of microglial InsR alters the phenotypic progression of obesity in both male and female mice. (**A**) Schematic representation of the Cre/loxP animal model used in this study. (**B**) Western blot and (**C**) quantification of InsR in primary microglia of InsR-Ctrl (*n* = 3) and InsR-KO (*n* = 3) mice, following 4-hydroxytamoxifen treatment. There was a significant decrease of InsR protein in InsR-KO microglia, compared to InsR-Ctrl microglia; (**D**,**E**) Female InsR-Ctrl and InsR-KO mice fed a Chow (black) or HFD (green) data for (**D**) food intake and (**E**) water intake during 12 weeks of diet exposure (*n* = 10–25); (**F**,**G**) Male InsR-Ctrl and InsR-KO mice fed a Chow (black) or HFD (blue) data for (**F**) food intake and (**G**) water intake during 12 weeks of diet exposure (*n* = 16–27). Female HFD-fed InsR-KOs had higher caloric intake after 3 weeks of HFD, compared to InsR-Ctrls, while male InsR-KO had lower caloric intake after 6 and 9 weeks of HFD, compared to InsR-Ctrls; (**H**,**I**) Female InsR-Ctrl and InsR-KO mice fed a Chow diet (black) or HFD (green) data for (**H**) body weight and (**I**) body weight gain presented in percentage, during 12 weeks of diet exposure (*n* = 10–25); (**J**,**K**) Male InsR-Ctrl and InsR-KO mice fed a Chow (black) or HFD (blue) data for (**J**) body weight and (**K**) body weight gain presented in percentage, during 12 weeks of diet exposure (*n* = 16–27); (**L**,**M**) Female InsR-Ctrl and InsR-KO mice fed a Chow diet (black) or HFD (green) data for (**L**) perirenal and (**M**) gonadal white adipose tissue (WAT) presented as percentage of body weight upon sacrifice at week 12 of diet exposure (*n* = 5–9). Perirenal WAT was higher in InsR-Ctrl HFD and InsR-KO Chow, compared to InsR-Ctrl Chow mice; (**N**,**O**) Male InsR-Ctrl and InsR-KO mice fed a Chow (black) or HFD (blue) data for (**N**) perirenal and (**O**) epididymal white adipose tissue presented as percentage of body weight upon sacrifice at week 12 of diet exposure (*n* = 4–12). Perirenal WAT was higher in Ins-KO compared to InsR-Ctrl male mice on the Chow diet. Data are presented as mean ± SEM *, #, $, & *p* < 0.05; **, ##, $$ *p* < 0.01; ***, ### *p* < 0.0001. Symbols used to show significance between groups in (D-K): * InsR-Ctrl: Chow vs. HFD; # InsR-KO: Chow vs. HFD; & Chow: InsR-Ctrl vs. InsR-KO; $ HFD: InsR-Ctrl vs. InsR-KO.

**Figure 2 ijms-23-02933-f002:**
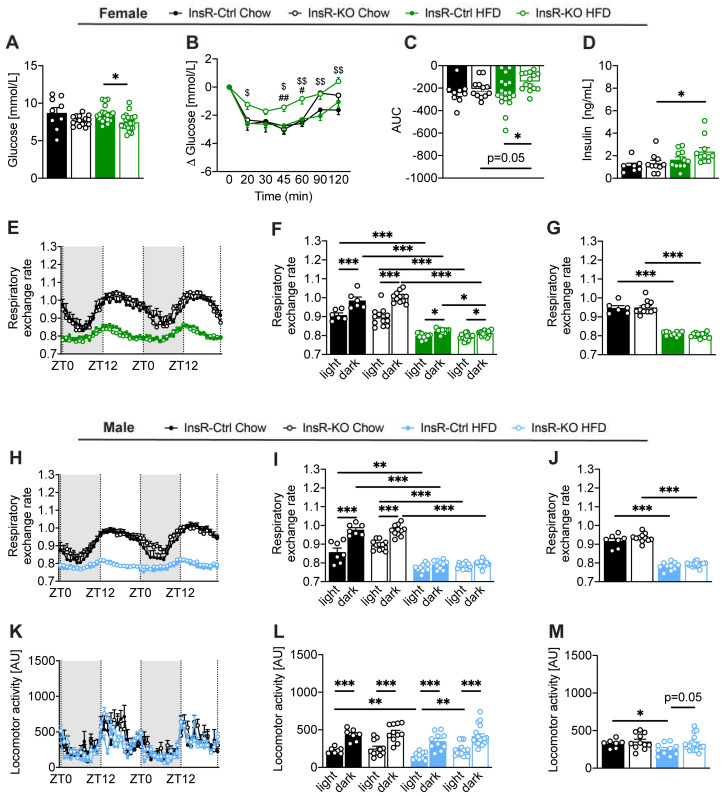
Loss of microglial InsR leads to impaired glucose homeostasis in obese female mice. (**A**–**D**) Female InsR-Ctrl and InsR-KO mice fed a Chow diet (black) or HFD (green) data for (**A**) fasted blood glucose, following 11 weeks of diet exposure (*n* = 9–19). InsR-KO female mice had lower blood glucose compared to InsR-Ctrl mice during obesity. (**B**,**C**) insulin tolerance glucose concentration presented as (**B**) delta glucose and (**C**) AUC data (*n* = 9–19). Delta glucose and AUC in InsR-KO females were higher compared to InsR-Ctrl mice during obesogenic conditions; (**D**) fasted plasma insulin, following 11–12 weeks of diet exposure (*n* = 7–11); (**E**–**G**) Female InsR-Ctrl and InsR-KO mice fed a Chow diet (black) or HFD (green) data for the respiratory exchange rate (RER) presented as (**E**) 48 h plot, (**F**) 48 h light and dark phase and (**G**) 24 h (*n* = 6–16). Loss of microglial InsR led to lower RER during the dark phase in obesogenic condition; (**H**–**J**) Male InsR-Ctrl and InsR-KO mice fed a Chow diet (black) or HFD (blue) data for respiratory exchange rate presented as (**H**) 48 h plot, (**I**) 48 h light and dark phase and (**J**) 24 h (*n* = 7–18). (**K**–**M**) Locomotor activity data presented as (**K**) 48 h plot, (**L**) 48 h light and dark phase, and (**M**) 24 h (*n* = 7–18). InsR-KO male mice had higher locomotor activity during the light phase compared to InsR-Ctrl mice during obesogenic conditions. RER and locomotor activity data are measured at approximately 11 weeks of diet exposure for the duration of 48 h. Data are presented as mean ± SEM. *, #, $ *p* < 0.05; **, ##, $$ *p* < 0.01; *** *p* < 0.0001. Symbols used to show significance between groups in (**B**): # InsR-KO: Chow vs. HFD; $ HFD: InsR-Ctrl vs. InsR-KO.

**Figure 3 ijms-23-02933-f003:**
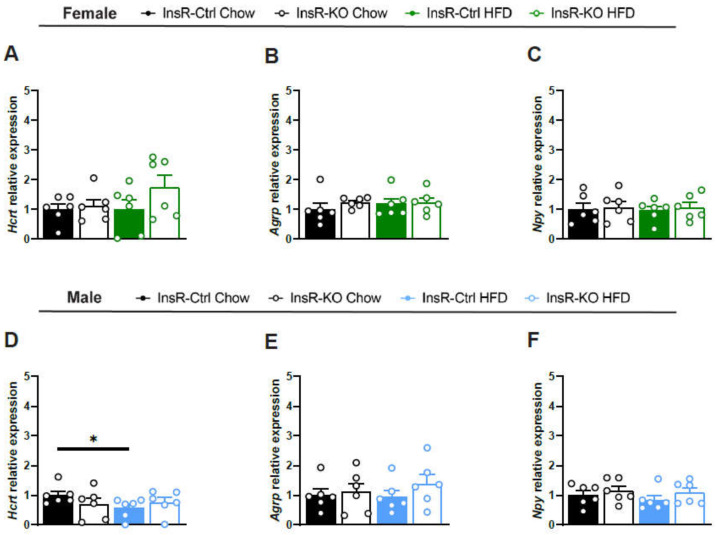
Loss of microglial InsR function has no effect on hypothalamic neuropeptide gene expression. (**A**–**F**) Relative gene expression of (**A**,**D**) *Hcrt*; (**B**,**E**) *Agrp* and (**C**,**F**) *Npy* from hypothalamic tissue There is a decrease in *Hcrt* expression in male InsR-Ctrl mice fed an HFD diet compared to Chow diet. Data presented for female InsR-Ctrl and InsR-KO mice fed a Chow diet (black), or HFD (green) and male InsR-Ctrl and InsR-KO mice fed a Chow diet (black) or HFD (blue), following 12 weeks of diet exposure (*n* = 6). Data are presented as mean ± SEM. * *p* < 0.05.

**Figure 4 ijms-23-02933-f004:**
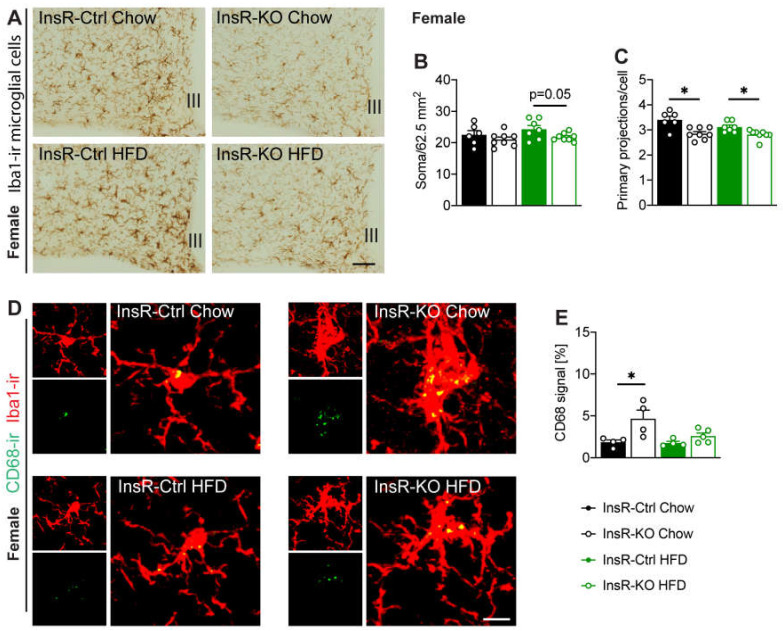
Lack of microglial InsR function leads to impaired microglial morphology and phagocytic capacity in the arcuate nucleus of female mice. (**A**) Iba1-ir microglia in the arcuate nucleus, Iba1-ir (**B**) soma number, and (**C**) primary projections (*n* = 6–8). There was a borderline significant decrease in microglial soma number in InsR-KO females, compared to InsR-Ctrl mice, when fed an obesogenic diet. InsR-KO mice had lower primary projections compared to InsR-Ctrl mice, irrespective of diet. (**D**,**E**) CD68-ir co-localization with Iba1-ir microglia in the ARC (*n* = 4–5). InsR-KO female mice had higher CD68-ir, compared to InsR-Ctrl mice under physiological conditions. Data presented for female InsR-Ctrl and InsR-KO mice fed a Chow diet (black) or HFD (green) following 12 weeks of diet exposure. Data are presented as mean ± SEM. * *p* < 0.05. III—3rd ventricle. Scale bar: 100 µm in (**A**); 10 µm in (**D**).

**Figure 5 ijms-23-02933-f005:**
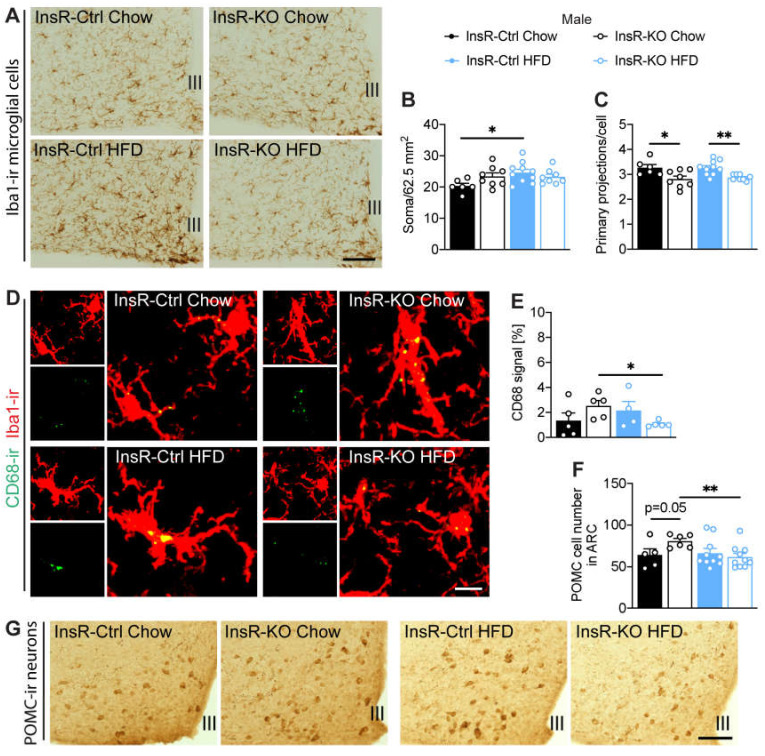
Loss of microglial InsR in male mice leads to impaired microglial morphology, altered phagocytic capacity, and changes in POMC cell number in the arcuate nucleus. (**A**) Iba1-ir microglia in the arcuate nucleus, Iba1-ir (**B**) soma number, and (**C**) primary projections (*n* = 6–10). InsR-KO mice had lower primary projections compared to InsR-Ctrl mice, irrespective of diet. (**D**,**E**) CD68-ir co-localization with Iba1-ir microglia in the ARC (*n* = 4–5). (**F**,**G**) POMC-ir neurons in the arcuate nucleus of male mice (*n* = 5–10). There was a borderline significant increase in POMC cell number in InsR-KO male mice, compared to InsR-Ctrl mice under physiological conditions. Data presented for male InsR-Ctrl and InsR-KO mice fed a Chow diet (black) or HFD (blue) following 12 weeks of diet exposure. Data are presented as mean ± SEM. * *p* < 0.05; ** *p* < 0.01. III—3rd ventricle. Scale bar: 100 µm in (**A**,**G**); 10 µm in (**D**).

## Data Availability

Data are contained within the article or Appendix A.

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
