# Peer review of "Loss of Microglial Insulin Receptor Leads to Sex-Dependent Metabolic Disorders in Obese Mice"

_ijms, 2022, doi:10.3390/ijms23062933_

Round 1
Reviewer 1 Report
The manuscript entitled “Loss of microglial insulin receptor leads to sex-dependent metabolic disorders in obese mice” by Milanova and coworkers evaluate very interesting topic, and I really enjoy reading
The manuscript is a very well-written and well organized, with adequate information on the subject. The applied methodology allows comprehensive insight to the topic. The data presented help to understand various variables and their relation to this topic.
I would suggest the author comment on the alterations in locomotor activity as a behavioral pattern usually employed as the manifestation of mood changes (such as anxiety and depression) in animal experimental models. Therefore, I can only propose a couple of papers that analyzed the impact of NPY and melanocortins on mood regulation with a number of parameters for estimation of locomotor activity in rats (https://doi.org/10.3389/fnins.2019.00172, https://doi.org/10.1371/journal.pone.0178922).
Delete a comment in Fig. 3C
Author Response
We thank the reviewer for the kind words and positive response to our work.
- I would suggest the author comment on the alterations in locomotor activity as a behavioral pattern usually employed as the manifestation of mood changes (such as anxiety and depression) in animal experimental models. Therefore, I can only propose a couple of papers that analyzed the impact of NPY and melanocortins on mood regulation with a number of parameters for estimation of locomotor activity in rats (https://doi.org/10.3389/fnins.2019.00172, https://doi.org/10.1371/journal.pone.0178922).
We thank the reviewer for this thoughtful suggestion. We have commented on the data accordingly in lines 187-189 in the revision (track changes).
- Delete a comment in Fig. 3C
We apologize for the inconvenience and have corrected the figure on page 8 in the revision.
Reviewer 2 Report
In the present work, the authors shows how the lack of microglial InsR in combination with high fat diet affect the homeostasis of glucose and microglial morphology. After reading such an excellent work, I can only thank the effort and work done by the authors. However, their work raises some doubts, that I would like to raise with the authors:
-Mayor considerations:
1) Check image 3C: it shows a text box that covers the graph
2) the authors must include the number of animals used in each experimental process and used for statistics (in the description of the figures)
3) The authors have induced a loss of microglial sensitivity to insulin (loss of its receptor), which has caused morphological changes and loss of glucose homeostasis capacity in microglial cells.
4) Have the authors observed morphological changes (atrophies) in any brain region or at the neuronal level (loss of synapses)?
5) Have the authors evaluated whether the observed microglial dysfunction has had repercussions on neuronal damage at the brain level? I think it would be interesting to evaluate the levels of phosphorylated tau, as a marker of brain/neuronal damage
6) I believe that the authors should include in their discussion these works in which the administration of intranasal insulin also promotes changes at the level of microglia:
DOI: 10.3389/fncel.2019.00188
doi: 10.3233/JAD-2012-121294.
DOI: 10.1186/s13024-017-0198-4
-Minor considerations:
I do not think that this point that I am going to comment on is totally necessary for this work, but I would like to convey to the authors my curiosity to know if their model presents any cognitive difficulty and I would like to know if the rate of neurogenesis has been altered. I think it is something very interesting that they could evaluate in future works, however if the authors have any information about it I think it would be very interesting to include it.
Author Response
In the present work, the authors shows how the lack of microglial InsR in combination with high fat diet affect the homeostasis of glucose and microglial morphology. After reading such an excellent work, I can only thank the effort and work done by the authors. However, their work raises some doubts, that I would like to raise with the authors:
We thank the reviewer for taking the time to read and critically evaluate our work. We are grateful for the kind words and positive response to our study. We have taken into consideration the suggestions of the reviewer and propose these changes.
-Major considerations:
1) Check image 3C: it shows a text box that covers the graph
We apologize for the inconvenience and have corrected the figure on page 8 in the revision.
2) the authors must include the number of animals used in each experimental process and used for statistics (in the description of the figures)
We have included the number of animals represented in each figure, the data can be found in the figure legends of Figures 1-5 (track changes).
3) The authors have induced a loss of microglial sensitivity to insulin (loss of its receptor), which has caused morphological changes and loss of glucose homeostasis capacity in microglial cells.
4) Have the authors observed morphological changes (atrophies) in any brain region or at the neuronal level (loss of synapses)?
We observed no brain atrophies or any differences in brain morphology between the different groups. In our previous experience with disruption of microglial function in physiological and obesogenic conditions, we haven’t observed obvious morphological changes to neuronal cells in the arcuate nucleus, apart from loss of POMC neurons (e.g. microglial deletion of lipoprotein lipase in mice led to accelerated loss of POMC neurons in the arcuate nucleus under obesogenic conditions (https://doi.org/10.1016/j.celrep.2017.09.008). In the current work, we chose to focus our attention on the arcuate nucleus due to its important role in the control of food intake and energy expenditure, as well as being a key region sensing changes in insulin function.
5) Have the authors evaluated whether the observed microglial dysfunction has had repercussions on neuronal damage at the brain level? I think it would be interesting to evaluate the levels of phosphorylated tau, as a marker of brain/neuronal damage
We thank the reviewer for this suggestion. Research from our group with rodent, as well as postmortem human brain tissue, have found hardly detectable phosphorylated tau in the hypothalamus including the arcuate nucleus (or infundibular nucleus in human). A previous study performed in postmortem patient brain tissue, evaluating tau pathology, showed a positive correlation with the progression of Alzheimer’s disease in nucleus basalis of Meynert (a non-hypothalamic area in the forebrain) (PMID: 26792551), but in our own study, we found no differences between control and diabetic patients in the NBM, nor in the hypothalamus (unpublished data), indicating even at the latest stage of T2DM, there is no detectable phosphorylated tau pathology in the hypothalamus. Moreover, the age of the mice used in this study was 12 weeks upon sacrifice, which is too early to observe any changes in ageing related pathology.
6) I believe that the authors should include in their discussion these works in which the administration of intranasal insulin also promotes changes at the level of microglia:
DOI: 10.3389/fncel.2019.00188
doi: 10.3233/JAD-2012-121294.
DOI: 10.1186/s13024-017-0198-4
We thank the reviewer for their suggestions. We have implemented them in the discussion section, lines 311-313 in the revision (track changes).
-Minor considerations:
I do not think that this point that I am going to comment on is totally necessary for this work, but I would like to convey to the authors my curiosity to know if their model presents any cognitive difficulty and I would like to know if the rate of neurogenesis has been altered. I think it is something very interesting that they could evaluate in future works, however if the authors have any information about it I think it would be very interesting to include it.
We agree with the reviewer that an evaluation of the impact of microglial insulin receptor deletion on cognition would be very interesting, however, at the time of performing the study we did not have the capacity to preform behavioral and cognitive evaluation of the animals. We did not observe “by eye” any major differences in behavior or cognition during the duration of the study.
Reviewer 3 Report
This manuscript by Milanova et al. evaluates the effects of a microglia-specific knockout of the insulin receptor gene in mice. The authors confirm knockout of insulin receptors (InsR) in the microglia of mice by culturing primary microglia from these mice. Male and female mice with or without the InsR knockout (InsR-KO or InsR-Ctrl, respectively) were fed either a control Chow diet or a high-fat diet to induce obesity. They characterized changes in progression of obesity, glucose homeostasis, neuropeptide gene expression, microglial morphology, and phagocytic capacity of microglia. The study finds that loss of insulin receptor in microglia leads to diet-dependent and sex-dependent changes in metabolism. The manuscript is generally clear and provides support for a role in microglia insulin receptor signaling in metabolism and important evidence of gender differences in experimental outcomes. Several changes could be made to improve clarity of the text and expand on interpretation of the results:
- Experiments have appropriate controls (chow diet vs. high-fat diet and control vs. insulin receptor knockout for male and female mice), but at times it is difficult to determine how the results of each comparison are being interpreted. For example, some differences seem to be depended on diet while others seem to be dependent on insulin receptor knockout. The authors should comment on this distinction to improve clarity.
- Given how variable diet and blood glucose levels can be, even between littermates a sample size of three seems very small for figure 1. There is also no indication of power analysis to determine the sample size needed to detect a significant difference.
- Although CD68 immunofluorescence is a proxy of the phagocytic function of microglia, a more functional experiment to test this would strengthen this claim. There are multiple assays that could be performed, such as testing the ability of microglia to engulf a fluorescently labeled particle.
- In the introduction, there is another study cited that found insulin decreased the innate immune response and altered the phagocytic capacity of microglia. The authors of this study report no direct effect on hypothalamic innate immunity. Including comments on this difference would add depth to the discussion.
- The final sentence in section 2.1 summarizes data suggesting that loss of microglial InsR plays a role in the progression of obesity in female mice, but the title of Figure 1 states that lack of microglial InsR alters progression of obesity in both male and female mice. This discrepancy should be clarified to avoid confusion.
- Under section 2.2, line 145 states that InsR-KO mice had higher basal blood glucose but there is no mention of the specific comparison made. Including that detail here would improve clarity.
- For Figure 4A and Figure 5A, a zoomed in image showing an individual cell (like the zoomed in images in panel D) would be informative. The zoomed in image could be used to show an example of cells with less primary projections in InsR-KO mice.
- In the methods section 4.2, were any known microglial markers used to confirm microglial identity in cultured cells before evaluating InsR protein levels?
- How were mice housed? Methods detail that mice were singly housed for RER, locomotor activity, and heat production measurements; were mice singly housed for all experiments? If mice were not singly housed, how was cumulative food and water intake recorded per animal?
- For insulin tolerance tests, is it standard in the field to use different doses for male and female mice? (0.75 U/kg for males vs. 0.5 U/kg for females)
- Minor concerns:
- Figure 3C is being partially obstructed by a text box. The title for Figure 3 has extra space after the word “gene”, and there is a period missing after the word “tissue” in the figure legend.
- Line 232 references “Figure 4C, 4C”, should reference 4C and 5C.
- Line 299: “mice” should be added after “male”
Author Response
We thank the reviewer for taking the time to read our work carefully and critically. We are grateful for the kind words. We have taken into consideration the suggestions of the reviewer and propose these changes/comments.
- Experiments have appropriate controls (chow diet vs. high-fat diet and control vs. insulin receptor knockout for male and female mice), but at times it is difficult to determine how the results of each comparison are being interpreted. For example, some differences seem to be depended on diet while others seem to be dependent on insulin receptor knockout. The authors should comment on this distinction to improve clarity.
The novelty of the study is that it presents a comparison between microglial-insulin receptor control and knockout animals. Where it was appropriate, we comment on the effects of diet in the context of previous available research or major findings.
- Given how variable diet and blood glucose levels can be, even between littermates a sample size of three seems very small for figure 1. There is also no indication of power analysis to determine the sample size needed to detect a significant difference.
The sample size of 3 concerns only 1C, where we perform cell culture experiments of pulled microglial cells from many neonates within the group. The revised manuscript contains the sample size for each panel/group of panels in the appropriate figure legend. We agree with the comment that high-fat diet (HFD) can induce variation and therefore our HFD-groups have higher sample size. All presented data excludes outliers. We have calculated sample size when applying for approval by the Netherlands institutions for performing this animal study for each experiment presented in this paper.
- Although CD68 immunofluorescence is a proxy of the phagocytic function of microglia, a more functional experiment to test this would strengthen this claim. There are multiple assays that could be performed, such as testing the ability of microglia to engulf a fluorescently labeled particle.
We have routinely performed these experiments in our group and observed a low phagocytic capacity of primary microglial cells when introduced to fluorescently labeled microspheres (unpublished data). These data often differ from observations in vivo, therefore, we chose not to perform these during this study and used the CD68 phagocytic indicator.
- In the introduction, there is another study cited that found insulin decreased the innate immune response and altered the phagocytic capacity of microglia. The authors of this study report no direct effect on hypothalamic innate immunity. Including comments on this difference would add depth to the discussion.
The study mentioned in the introduction was performed using the BV2 murine microglial cell line, therefore, the study was performed only in vitro and cannot provide any information about in vivo brain region or sex differences. That is why we chose not to address it in the discussion.
- The final sentence in section 2.1 summarizes data suggesting that loss of microglial InsR plays a role in the progression of obesity in female mice, but the title of Figure 1 states that lack of microglial InsR alters progression of obesity in both male and female mice. This discrepancy should be clarified to avoid confusion.
We thank the reviewer for noticing. We have clarified the conclusion in section 2.1, line 108 in the revision (track changes).
- Under section 2.2, line 145 states that InsR-KO mice had higher basal blood glucose but there is no mention of the specific comparison made. Including that detail here would improve clarity.
We have clarified this in line 146 in the revision (track changes).
- In the methods section 4.2, were any known microglial markers used to confirm microglial identity in cultured cells before evaluating InsR protein levels?
We have performed neonatal primary microglial cell isolation routinely in our group and at the beginning of protocol optimization, we evaluated the purity of the collected microglial cells, testing for different cell markers. We found no contamination for the main cell types that could survive the culture process (not published).
- How were mice housed? Methods detail that mice were singly housed for RER, locomotor activity, and heat production measurements; were mice singly housed for all experiments? If mice were not singly housed, how was cumulative food and water intake recorded per animal?
As mentioned in line 358 of the manuscript, all mice were group housed during the duration of the study, prior to metabolic cage experiments. From there onwards, the animals were single housed up until sacrifice.
- For insulin tolerance tests, is it standard in the field to use different doses for male and female mice? (0.75 U/kg for males vs. 0.5 U/kg for females)
Prior to the start of the insulin tolerance testing, we performed a literature search for optimal concentration. 0.75 U/kg is a standard dose used for male mice and we found only one publication at the time that mentioned females and the necessity to optimize. We performed a pilot with all groups with 0.75 U/kg in both males and females, which lead to a severe hypoglycemia in many females (within all experimental groups) and we terminated prior to the end of the test. Therefore, we reduced the dose to 0.5 U/kg, where we could perform comparable experiments in female mice.
Minor concerns:
- Figure 3C is being partially obstructed by a text box. The title for Figure 3 has extra space after the word “gene”, and there is a period missing after the word “tissue” in the figure legend.
We apologize for the inconvenience and have uploaded the correct figure in the revised manuscript.
- Line 232 references “Figure 4C, 4C”, should reference 4C and 5C. We thank the reviewer for this observation.
We have corrected it in line 236 in the revision (track changes).
- Line 299: “mice” should be added after “male”
We thank the reviewer for this observation. We have corrected it in line 303 in the revision (track changes).
Round 2
Reviewer 2 Report
After reviewing this document again, I thank the authors for the changes and explanations provided on the manuscript. I think it is a very interesting work, in addition the authors have convincingly explained each of the doubts raised above.